# Goat’s Milk Intake Prevents Obesity, Hepatic Steatosis and Insulin Resistance in Mice Fed A High-Fat Diet by Reducing Inflammatory Markers and Increasing Energy Expenditure and Mitochondrial Content in Skeletal Muscle

**DOI:** 10.3390/ijms21155530

**Published:** 2020-08-01

**Authors:** Claudia Delgadillo-Puga, Lilia G. Noriega, Aurora M. Morales-Romero, Antonio Nieto-Camacho, Omar Granados-Portillo, Leonardo A. Rodríguez-López, Gabriela Alemán, Janette Furuzawa-Carballeda, Armando R. Tovar, Luis Cisneros-Zevallos, Ivan Torre-Villalvazo

**Affiliations:** 1Departamento de Nutrición Animal Dr. Fernando Pérez-Gil Romo, Instituto Nacional de Ciencias Médicas y Nutrición Salvador Zubirán (INCMNSZ), Ciudad de Mexico 14080, Mexico; 2Departamento de Fisiología de la Nutrición, Instituto Nacional de Ciencias Médicas y Nutrición Salvador Zubirán (INCMNSZ), Ciudad de Mexico 14080, Mexico; lgnoriegal@gmail.com (L.G.N.); ograpo@yahoo.com (O.G.-P.); akumakuja@hotmail.com (L.A.R.-L.); galemane@gmail.com (G.A.); armando.tovarp@incmnsz.mx (A.R.T.); 3Facultad de Química, Universidad Nacional Autónoma de México (UNAM), Ciudad de Mexico 04510, Mexico; aurizmorales9@gmail.com; 4Instituto de Química, Universidad Nacional Autónoma de México (UNAM), Ciudad de Mexico 04510, Mexico; camanico2015@gmail.com; 5Departamento de Inmunología y Reumatología, Instituto Nacional de Ciencias Médicas y Nutrición Salvador Zubirán (INCMNSZ), Ciudad de Mexico 14080, Mexico; jfuruzawa@gmail.com; 6Department of Horticultural Sciences, Texas A&M University, College Station, TX 77843-2133, USA; lcisnero@tamu.edu; 7Department of Nutrition and Food Science, Texas A&M University, College Station, TX 77843, USA

**Keywords:** goat’s milk, polyphenols, adipose tissue browning, functional food, bioactive compounds, energy expenditure, grazing, *Acacia farnesiana*

## Abstract

Goat’s milk is a rich source of bioactive compounds (peptides, conjugated linoleic acid, short chain fatty acids, monounsaturated and polyunsaturated fatty acids, polyphenols such as phytoestrogens and minerals among others) that exert important health benefits. However, goat’s milk composition depends on the type of food provided to the animal and thus, the abundance of bioactive compounds in milk depends on the dietary sources of the goat feed. The metabolic impact of goat milk rich in bioactive compounds during metabolic challenges such as a high-fat (HF) diet has not been explored. Thus, we evaluated the effect of milk from goats fed a conventional diet, a conventional diet supplemented with 30% *Acacia farnesiana* (AF) pods or grazing on metabolic alterations in mice fed a HF diet. Interestingly, the incorporation of goat’s milk in the diet decreased body weight and body fat mass, improved glucose tolerance, prevented adipose tissue hypertrophy and hepatic steatosis in mice fed a HF diet. These effects were associated with an increase in energy expenditure, augmented oxidative fibers in skeletal muscle, and reduced inflammatory markers. Consequently, goat’s milk can be considered a non-pharmacologic strategy to improve the metabolic alterations induced by a HF diet. Using the body surface area normalization method gave a conversion equivalent daily human intake dose of 1.4 to 2.8 glasses (250 mL per glass/day) of fresh goat milk for an adult of 60 kg, which can be used as reference for future clinical studies.

## 1. Introduction

Whole milk is a significant source of nutrients, including fat, protein, minerals and vitamins. Besides its high nutritional value, whole milk contains several bioactive compounds (peptides, conjugated linoleic acid, short chain fatty acids, monounsaturated and polyunsaturated fatty acids and polyphenols including isoflavones, among others) with important beneficial health effects. Thus, whole milk has been recognized as a functional food [1,2]. Despite this evidence, there is a prevalent notion that foods with elevated fat content; including whole milk, induces weight gain and are a primary cause of obesity; particularly because of their elevated content of saturated fat. Saturated fat has been associated with increased low-density lipoprotein cholesterol levels, which in turn increase the risk of cardiovascular disease (CVD). For this reason, there is an increasing trend to replace natural high-fat dairy foods for low-fat or non-fat dairy products considered by consumers as “healthier alternatives”. However, recent research reveals that intake of high-fat milk and dairy products have no association with the occurrence of CVD, stroke, coronary heart disease, dyslipidemia or type 2 diabetes [3,4]. Moreover, milk intake provides health benefits in subjects with diabetes [5], obesity [6] and metabolic syndrome [7] in particular with fermented dairy [8]. Despite this body of evidence, the consumption of dairy products remains a subject of debate.

The variation in the beneficial effects of milk from different studies may arise from the huge number of variables that impacts the composition of milk, including the animal species from which it comes, composition of feed and even the animal care and welfare [9]. The majority of studies involving dairy products have been conducted using cow’s milk. We and others have shown that goat’s milk is also a rich source of nutrients and bioactive compounds [10]. However, there is scarce evidence regarding the effect of goat’s milk on health [11]. We have previously demonstrated that goat’s milk nutrient composition and bioactive compounds content varies depending on the composition of dairy goat’s feed [12]. Thus, the handling and variety of food provided to goats (grazing, organic feeding, strategic supplementation, etc.) determines the quality of milk [10]. Accordingly, we have demonstrated that in dairy goats, grazing management increases the content of polyunsaturated fatty acids of milk with respect to goats fed a conventional diet. Moreover, inclusion of *Acacia farnesiana* (AF) pods boosted the polyphenol content and increased the *n*−3:*n*−6 fatty acids ratio [10], improving the quality and functional potential of milk. Nevertheless, whether the different quality of the milk is reflected on health benefits during metabolic challenges such as a high-fat (HF) diet has not been explored. Thus, the objective of this study was to determine the effect of feeding lyophilized whole milk from goats fed conventional diet (CD), grazing or CD supplemented with AF pods on the inflammatory and metabolic parameters of mice fed a HF diet.

## 2. Results

### 2.1. Goat’s Milk Intake Attenuates Body Weight Gain, Increases Food Intake, and Modifies Body Composition and Serum Parameters in Mice Fed A High-Fat Diet

To evaluate the effect of goat milk intake in the development of obesity and its metabolic consequences, we fed mice with a control diet (control), a high fat diet (HF) or a HF diet supplemented with lyophilized milk from goats fed a conventional diet (HFCD), grazing (HFG) or a conventional diet supplemented with *Acacia farnesiana* pods (HFAF) for 14 weeks. As expected, at the end of the study, mice fed the HF diet showed a significant increase (*p* < 0.05) in body weight with respect to those fed the control diet. Interestingly, the three groups of mice fed HF diet supplemented with goat milk (HFCD, HFG, and HFAF) presented similar body weight to those fed the control diet throughout the study period (Figure 1A,B). The lower final body weight of mice fed goat milk with respect to those fed the HF was not caused by a reduction in food or energy intake. As observed in Figure 1C, mice fed HFCD, HFG or HFAF had higher food intake than those fed the HF. Since HF diets are more energy-dense than the control diet, energy intake of mice fed either diet containing goat milk was higher than those fed control or HF diets (*p* < 0.05) (Figure 1D). The increased body weight of HF mice was due to a significant increment in body fat mass (*p* < 0.05) and reduced lean mass with respect to control (Figure 1E,F). Notably, mice fed goat’s milk presented no significant differences in fat and lean mass with respect to control except the HFCD group that had lower fat mass and higher lean mass than all other groups. Leptin is an adipose-derived hormone that is secreted in proportion to total body fat [13]. As expected, circulating leptin concentration was directly associated with fat mass in all groups (Figure 1G). To evaluate the effect of dietary goat milk in circulating lipids we determined serum triglycerides and cholesterol. As expected, HF mice presented higher serum triglycerides than control mice (*p* < 0.05). Nevertheless, HFCD, HFG, and HFAF mice presented significantly lower triglycerides levels than HF mice (*p* < 0.05), similar to those of control mice (Figure 1H). 

High-fat diet feeding did not increase serum cholesterol with respect to control mice. Only the HFG and HFAF diets presented a modest but significant increase in serum cholesterol with respect to all other groups (Figure 1I). However, neither group presented cholesterol concentration above reference values for C57BL/6 mice (130 ± 10) [14]. These results indicate that consumption of goat milk prevents fat accretion and body weight gain and decreases serum leptin and triglycerides levels of mice fed a HF diet despite increased energy intake.

### 2.2. Goat’s Milk Consumption Prevents Insulin Resistance and Pancreatic Islets Hypertrophy in Mice Fed A High-Fat Diet

Chronic consumption of a high-fat diet induces metabolic derangements such as glucose intolerance and insulin resistance [15]. At the end of the study, fasting serum glucose was higher in the HF mice when compared to controls and mice fed HFCD (*p* < 0.05) (Figure 2A). Mice fed HFG and HFAF had no significant difference in fasting glucose concentration with respect to HF or control groups. However, fasting serum insulin at the end of the study was significantly lower in all groups fed goat milk than in those fed the HF or control diets (Figure 2B). To determine whether goat milk modifies glucose homeostasis in mice fed a HF diet, we performed an intraperitoneal glucose tolerance test (IpGTT). As expected, mice fed HF presented a significant (*p* < 0.05) decrease in glucose tolerance as observed by the higher area under the curve (AUC) when compared to control mice (Figure 2C,D).

Interestingly, mice fed with either goat’s milk had the same glucose tolerance than control. Then, to examine whether the improved glucose tolerance observed in animals fed goat milk was associated with higher insulin sensitivity, we performed an intraperitoneal insulin tolerance test (ipITT). As expected, mice fed HF had a blunted response to exogenous insulin reflected by a higher AUC (Figure 2E,F). 

Interestingly, the glucose curve of mice fed goat’s milk during the ipITT challenge was lower than those fed HF and similar to that of control mice. The AUC of all groups fed goat milk had a trend to be lower than that of the HF, but only in HFAF was significantly lower than the other groups. The development of insulin resistance is accompanied with a parallel increase in insulin secretion in order to maintain euglycemia. Moreover, glucose and insulin are trophic factors that induces islet compensatory growth response to insulin resistance. Thus, persistently elevated glucose and insulin concentrations induces hypertrophy of pancreatic islets [16]. 

Consistently with the aforementioned results, pancreatic islets size was higher in mice fed the HF diet than control, HFCD and HFAF (*p* < 0.05) (Figure 2G,H). These results indicate that goat’s milk decreases HF-associated glucose intolerance, preventing hyperinsulinemia. Notably, the lower insulin concentration of mice fed goat milk was associated with a decrease in HF-induced pancreatic islet hypertrophy, since HFCD, HFG and HFAF mice presented pancreatic islets size similar to those from control mice. Altogether, these results indicate that goat’s milk intake attenuates glucose intolerance and hyperinsulinemia associated to a decrease in pancreatic islet size in mice fed a HF diet.

### 2.3. Goat’s Milk Consumption Increases Whole-Body Energy Expenditure and Attenuates Metabolic Inflexibility by an Increase in UCP-1 in BAT and Oxidative Fibers in Muscle of Mice Fed A High-Fat Diet

To evaluate whether the lower body weight and fat mass, and the improvement on glucose tolerance observed in HF mice fed goat’s milk was associated with increased energy expenditure, we evaluated oxygen consumption by indirect calorimetry using a CLAMS system. As expected, HF mice had lower average oxygen consumption than control mice in both fasting and feeding conditions (Figure 3A,B). Interestingly, HFCD, HFG and HFAF mice presented significantly higher oxygen consumption than HF and control mice in fed conditions (Figure 3A,B), indicating an increase in energy expenditure. When we calculated oxygen consumption per Kg of lean mass, only de HFAF group had higher oxygen consumption than the rest of the groups (Figure 3C). In addition, a linear regression analysis showed that the slopes of change of VO_2_ (mL/h) per unit change of body weight (Figure 3D) or lean mass (Figure 3E) were similar between control, HF and HFCD groups but not for the HFG and HFAF groups. Moreover, when we evaluated the effect of goat’s milk on substrate utilization by calculating the respiratory exchange ratio (RER), control mice had a RER of 0.8 during fasting conditions (Figure 3F,G), which increased to 1.0 during the fed state indicating an adequate metabolic flexibility, that is the ability to change from fatty acids to glucose utilization when substrates become available during the feeding period. Conversely, the RER of HF mice remained close to 0.8 during the feeding period (Figure 3F,G), indicating the presence of metabolic inflexibility or the inability to change substrate utilization. HFCD, HFG, and HFAF mice presented a significantly higher RER than HF mice, indicating an improvement on substrate utilization. The lower RER of mice fed goat’s milk with respect to those fed the control diet indicates that these mice uses fatty acids as an energy substrate even during the feeding period. However, since these mice also presented increased energy expenditure, these results indicate that mice fed goat’s milk had augmented fat oxidation capacity with respect to those fed HF, preventing excessive fat deposition in adipose tissue or other organs.

Maintenance of a stable body weight despite a positive energy balance in response to consumption of an energy-dense diet is achieved by increased energy expenditure. Non-shivering thermogenesis in brown adipose tissue (BAT) and mitochondrial oxidation in skeletal muscle are two high energy-demanding activities that increases energy expenditure [17]. To evaluate whether the increase on energy expenditure was associated with BAT remodeling and skeletal muscle mitochondrial content, we evaluated BAT morphology and lipid content and mitochondrial activity in skeletal muscle. BAT of mice fed the HF diet contained a majority of adipocytes with one single large lipid vacuole, in contrast with the BAT of control mice, composed of multilocular adipocytes with small lipid vacuoles (Figure 4A). The increase in lipid vacuoles size of brown adipocytes is named “whitening” and reflects an impaired capacity to oxidize substrates and thus, reduced thermogenic activity [18]. The mean vacuole size in brown adipocytes from mice fed the HF diet was significantly higher with respect to controls (Figure 4B). Interestingly, the mean lipid vacuoles size of BAT from mice fed goat’s milk were similar than that of the control mice. To evaluate if the reduced lipid content in BAT was associated with increased thermogenesis, we evaluated uncoupling protein 1 (UCP-1) abundance in BAT by immunohistochemistry. Notably, mice fed with either goat’s milk had higher expression (*p* < 0.05) of UCP-1 with respect to mice fed the HF diet (Figure 4A,C). UCP-1 expression was also evaluated by western blot. UCP-1 protein abundance was higher (*p* < 0.05) in BAT of mice fed the HFG and HFAF than in mice fed the HF diet (Figure 4D,E). With regard to skeletal muscle, BODIPY staining showed increased lipid content in muscle fibers of mice fed HF with respect to control (Figure 4F). Interestingly, lipid accumulation in muscle fibers of mice fed with HFCD, HFG and HFAF diets was significant (*p* < 0.05) lower than those fed HF as determined by densitometric quantification of BODIPY staining (Figure 4G). Succinate dehydrogenase (SDH) staining is a measure of mitochondrial oxidative capacity. As expected, mice fed HF had significantly lower oxidative capacity than those fed the control diet (Figure 4F). Interestingly, mice fed with HFCD, HFG and HFAF diets presented significantly higher SDH enzyme activity in skeletal muscle than those fed HF and even higher than control (*p* < 0.05) (Figure 4H). Mitochondrial biogenesis and activity in skeletal muscle is regulated by the combined action of the cellular energy sensor AMP-activated protein kinase (AMPK) and the nuclear receptor peroxisome proliferator-activated receptor delta (PPARδ) [19]. Interestingly, AMPK phosphorylation significantly increased in skeletal muscle of HFCD, HFG and HFAF when compared to HF and control mice (Figure 4I,J) despite no significant difference in the mRNA abundance of AMPK between groups (Figure 4K). Messenger RNA abundance of PPARδ in skeletal muscle was higher (*p* < 0.05) in HFG and HFAF compared with the rest of dietary treatments (Figure 4L). These results indicate that goat’s milk intake increases oxidative metabolism in skeletal muscle and substrate oxidation in BAT, preventing body fat accumulation and body weight gain despite higher energy intake with respect to control or the HF group.

### 2.4. Goat’s Milk Intake Prevent White Adipose Tissue Hypertrophy and Modulates Adipokine mRNA in Adipocytes of Mice Fed A High-Fat Diet

Chronic intake of high-energy diets induces adipose tissue expansion in order to efficiently store energy excess as triglycerides, preventing excessive lipid accumulation in other organs. However, macrophage activation in adipose tissue induces a local inflammatory response leading to a reduction in adipocyte proliferation and hypertrophy of existing adipocytes [20]. Histological analysis of subcutaneous and visceral adipose tissue sections revealed that the HF diet significantly increased mean adipocyte size in both adipose tissue depots with respect to control (Figure 5A,B,H). Despite the high-fat diets added with goat’s milk, were isoenergetic with respect to the HF diet; subcutaneous and visceral adipose tissues of mice fed goat’s milk presented adipocytes significantly smaller than those fed HF. Frequency distribution analysis revealed that 50% of adipocytes from control mice were smaller than 1000 µm whereas less than 25% of adipocytes from HF mice were in that size range (Figure 5C). Interestingly, as in control, more than 40% of adipocytes of mice fed HFCD, HFG or HFAF were smaller than 1000 µm. In visceral adipose tissue of HF mice, less than 35% of adipocytes were under 1500 µm, whereas in control mice and those fed goat’s milk more than 50% of adipocytes were under that range (Figure 5I). To determine if the smaller size of adipocytes from mice fed goat’s milk was associated with increased fatty acid lipolysis and thermogenesis we evaluated UCP-1 distribution and the activity of hormone sensitive lipase (HSL) in subcutaneous adipose tissue. UCP-1 expression in subcutaneous adipose tissue of all groups fed goat’s milk was higher with respect to the group fed HF (Figure 5A,D). The activity of HSL in all groups fed goat’s milk was higher (*p* < 0.05) respect to control and HF diets (Figure 5E,F). These results indicate that goat’s milk intake increases the hydrolysis of triglycerides in white adipose tissue to sustain in situ UCP-1 mediated thermogenesis, preventing adipocyte hypertrophy [21]. As expected, a response to reduced adipocyte size is the reduction in leptin mRNA expression. Leptin mRNA content in HFCD, HFG and HFAF was higher than in control but significantly lower than in HF (Figure 5J). These results indicate that goat’s milk favors adipose tissue expansion during chronic energy intake, preventing adipocyte hypertrophy. Adipose tissue expandability in response to positive energy balance is mediated by the activity of the adipose-tissue specific transcription factor PPARγ2 which in turn stimulates adiponectin gene expression and secretion [22]. The mRNA abundance of PPARγ2 was significantly higher (*p* < 0.05) in subcutaneous adipose tissue of mice fed HFCD, HFG or HFAF with respect to those fed control or HF (Figure 5K). Accordingly, adiponectin mRNA content in adipose tissue was also higher in subcutaneous adipose tissue of mice fed goat’s milk than in those fed control or HF diet (Figure 5L). On the contrary, hypertrophic adipose tissue is characterized by macrophage recruitment and increased cytokine secretion [20]. Immunohistochemistry of the macrophage marker F4780 revealed numerous macrophage infiltrates in adipose tissue of mice fed HF (Figure 5A). No positive staining was detected in adipose tissue sections of mice fed control or goat’s milk. However, tumor necrosis factor α content in adipose tissue homogenates was not significantly between diets (Figure 5M). These results indicate that goat’s milk intake can activate the nuclear receptor PPARγ2, increasing adiponectin gene expression preventing infiltration of macrophages in mice fed a HF diet.

### 2.5. The Anti-Inflammatory Fatty Acid Profile of Goat Milk’s Prevents Hepatic Lipid Deposition and Inflammatory Markers and Increases AMPK Phosphorylation

Excessive lipid deposition in liver induces cytotoxic cell death and inflammation, leading to the progression of non-alcoholic fatty liver disease to steatohepatitis. However, dietary fatty acid composition can modify hepatic lipid metabolism and the inflammatory response. Long-chain saturated fatty acids such as stearic acid favors lipid accumulation and inflammation whereas long-chain *n*-3 polyunsaturated fatty acids such as eicosapentaenoic (EPA), docosahexaenoic (DHA) and rumen bacteria-derived conjugated linoleic acid (CLA) exerts anti-inflammatory activities in liver [23].

To evaluate if the fatty acid profile in liver reflects that of experimental diets, we determined the fatty acid profile of experimental diets and liver. The most abundant saturated fatty acids in goat milk are the short-chain myristic (C14:0), capric (C8:0) and the long-chain stearic (C18:0). Regardless of the origin of the goat milk, all diets containing goat’s milk showed a greater amount of myristic and capric acids with respect to control or HF (Figure 6A,B). Moreover, the amount of stearic acid in diets containing goat’s milk was lower compared to the HF diet (Figure 6C). This pattern was reflected in the fatty acids present in the liver (Figure 6A–C). The level of the pro-inflammatory fatty acid arachidonic acid was lower in the diets containing goat’s milk than in HF or control (Figure 6D). Hepatic arachidonic acid levels presented the same pattern. Moreover, the levels of EPA and CLA were higher in the diets that contains goat milk with respect to HF or control (Figure 6E,F), which was reflected in the hepatic content of these fatty acids. These results indicate that hepatic fatty acid composition reflects the composition of dietary fatty acids and suggest that goat’s milk fatty acid profile could exert beneficial effects in liver such as reduced lipid accumulation and lower inflammation.

The primary source of hepatic fat due to calorie excess is de novo lipogenesis mediated by the activity of the sterol regulatory element binding protein-1c (SREBP-1c). This transcription factor mediates gene expression of lipogenic enzymes in the liver [24]. To evaluate the effect of goat milk intake on the development of fatty liver and SREBP-1c expression, H&E and ORO staining and real-time PCR quantification of SREBP-1c mRNA were performed on liver samples from the different experimental groups. As expected, mice fed HF presented a high number of hepatocytes containing enlarged lipid vacuoles (Figure 6G). Accordingly, densitometric quantification of ORO staining revealed a significant increase in lipid content of liver of HF mice with respect to control (Figure 6H). Notably, mice fed with the HFCD, HFG and HFAF diets showed less fat accumulation in liver and cytostructural features similar to those of control mice. Messenger RNA abundance of SREBP-1c was higher in the liver of mice fed HF with respect to control (Figure 6I). Strikingly, SREBP-1c gene expression was lower in the liver of animals fed with HFCD, HFG and HFAF diets with respect to HF. The expression of SREBP-1c in liver increases in response to hyperinsulinemia/insulin resistance but is reduced by particular fatty acids such as EPA and DHA [25]. Accordingly, SREBP-1c mRNA abundance in liver was inversely correlated with hepatic DHA content (Figure 6J). Another beneficial effect of the fatty acids presents in goat’s milk, such as EPA is the activation of AMPK, the cellular energy sensor [26]. Upon activation, AMPK increases glucose and fatty acid oxidation, reducing lipid content in liver [27]. As observed in Figure 6K,L, HFG and HFAF increased AMPK phosphorylation in liver with respect to the rest of the experimental diets (Figure 6K). These results indicate that consumption of goat’s milk in mice fed a HF diet prevents hepatic steatosis in part by reducing SREBP-1c-mediated lipogenesis and, simultaneously, by increasing AMPK-mediated lipid oxidation.

Besides the increase in lipogenesis, an increase in hepatic oxidative and endoplasmic reticulum stress are determining factors in the development of non-alcoholic fatty liver disease by activating JNK phosphorylation, a common downstream element in oxidative and endoplasmic reticulum stress signaling pathways [28]. As observed in Figure 6K,M, the abundance of phosphorylated JNK was higher in liver of mice fed HF with respect to control. Notably, HFCD, HFG and HFAF fed mice presented lower levels of p-JNK than those observed in HF. To evaluate if the reduced JNK phosphorylation was associated with lesser intrahepatic TNF-α content, we determined this cytokine in liver homogenates. Accordingly, TNF-α abundance was higher in liver homogenates of HF mice with respect to control, HFCD, HFG and HFAF (Figure 6N). 

Finally, we determined the hepatic EPA+DHA/AA ratio. This ratio is used as an anti-inflammatory index, where a higher ratio indicates a protective role in the development of inflammatory states [29]. Notably, mice fed goat’s milk presented a significant higher hepatic EPA+DHA/AA ratio with respect to those fed HF or control (Figure 6O). These results indicate that goat’s milk intake prevents hepatic inflammation in mice fed a HF associated to a protective proportion of *n*-3/*n*-6 long chain polyunsaturated fatty acids in liver.

## 3. Discussion

The present work demonstrates that the incorporation of dry whole goat’s milk to a high-fat (HF) diet prevents excessive body weight gain, body fat mass accretion and hepatic steatosis in mice. These beneficial effects were elicited through increased whole-body insulin sensitivity and energy expenditure, improving substrate utilization during the feeding-fasting cycles. The metabolic effects of goat’s milk intake could be attributed to an increase in oxidative fibers in skeletal muscle, augmented UCP1 expression in brown adipose tissue, a reduction in SREBP-1c mediated lipogenesis in liver and increased adipose tissue browning, lipolysis and *in situ* thermogenesis.

The beneficial effects of goat’s milk on the metabolic profile of mice, despite chronic consumption of a high-energy diet, indicates that goat’s milk contains particular molecules that can modulate metabolic and inflammatory pathways in different organs. Goat’s milk is a significant source of plant-derived phenolic compounds. We and other have previously reported the polyphenol composition of goat’s milk [10,30,31]. We also observed that goat fed influences polyphenol content in milk. Goat fed supplemented with 30% of *Acacia farnesiana* pods or grazing increases the polyphenol content in goat’s milk by approximately 2-fold and 1.4-fold, respectively [10]. This differential polyphenol enrichment was reflected in the metabolic effects induced by the different diets.

Among the polyphenols found in goat’s milk, gallic acid, chlorogenic acid, ferulic acid and catechins are the most abundant [10], which have demonstrated to have antioxidant properties [32,33,34], increase in energy expenditure in mice [35] and humans [36] and anti-inflammatory properties [37]. Thus, the polyphenol content in goat’s milk could be directly responsible of the increase in energy expenditure. We have previously demonstrated that polyphenols such as genistein increases energy expenditure in mice through modulation of AMPK and the thermogenic program of subcutaneous adipose tissue [38]. The activation of AMPK in skeletal muscle and liver of mice fed goat’s milk was associated with lower lipid content in both tissues, increased mitochondrial content in skeletal muscle and whole-body glucose tolerance. Thus, the polyphenol content of goat’s milk could exert these metabolic activities by the stimulation of AMPK activity.

Moreover, the effect of polyphenols in metabolic tissues could also be attributed to the activation of metabolic nuclear receptors [39]. For instance, nuclear receptor PPARγ2 is the master regulator of adipose tissue functions, controlling adipocyte differentiation, lipid storage, adipokine secretion and immune activities of resident macrophages in adipose tissue [40]; PPARδ governs substrate utilization and oxidative metabolism in skeletal muscle by activating the transcriptional program for mitochondrial biogenesis [41]. Interestingly, we found a significant increase in PPARγ2 expression in adipose tissue, and PPARδ in skeletal muscle of mice fed goat’s milk, suggesting that some of the polyphenols present in goat’s milk are able to activate these nuclear receptors in metabolic tissues. The reduction in adipocyte hypertrophy, increased UCP-1 and adiponectin expression and reduced macrophage infiltration in adipose tissue was likely due to activation of PPARγ2 by goat’s milk polyphenols. Similarly, the increase in mitochondrial activity and reduction of lipid deposition in skeletal muscle fibers was probably result of the combined action of AMPK activation, and PPARδ transcriptional activity.

Another type of molecules in goat’s milk which are able to modulate metabolism and inflammation are fatty acids. The consumption of specific monounsaturated (MUFA) or polyunsaturated (PUFA) fatty acids can exert beneficial metabolic effects. The 16:1n-7 and 18:1n-9 MUFAS and 20:5(*n*-3) and 22:6(*n*-3) PUFAS are known to promote blood pressure control, adequate coagulation, enhanced endothelial function and preservation of insulin sensitivity, having beneficial effects on the prevention and treatment of metabolic syndrome [41]. MUFA-rich diets can improve glycemic control while diets rich in *n*-3 PUFA such as EPA and DHA can improve plasma TG levels, and favor the resolution of inflammation [42,43]; whereas high intake of the *n*-6 PUFA arachidonic acid (AA) induces low-grade inflammation, oxidative stress, endothelial dysfunction and atherosclerosis [44]. According to this scenario, the lower body fat mass (Figure 1E), the higher insulin sensitivity (Figure 2E), the lower fatty acid accumulation on liver (Figure 6G,H), and finally, the input of *n*-3 PUFA anti-inflammation (Figure 6O) of goat’s milk prevented the establishment of metabolic syndrome in our mice as showed our results.

Interestingly, it has been demonstrated that the substitution of saturated with monounsaturated fatty acid attenuates hyperinsulinemia and pancreatic islet dysfunction [45]. Thus, it is possible that the specific fatty acid profile of goat’s milk could be responsible (Figure 6D–F) of the decrease in hyperinsulinemia and pancreatic islet hypertrophy observed in mice fed these diets (Figure 2E,G,H), which is in contrast with lard that has a high content in saturated long fatty acids. In fact, deletion of Elov16 enzyme that elongates fatty acids protects pancreatic islets from the hypertrophy induced by high-fat diets [46]. Moreover, the high content of EPA, ALA and DHA, could modulate insulin secretion by a G-protein-receptor-mediated release of glucagon-like peptide 1 (GLP-1) by enteroendocrine L-cells [47], and insulin sensitivity pathways in other tissues [48,49], decreasing the stress on pancreatic islets and, therefore, pancreatic islets hypertrophy.

We cannot discard other mechanisms that may involve other components through which goat’s milk exert its beneficial effect, such as micro RNAs contained in exosomes [50], or bioactive peptides [51]. Exosomes have been found in the skim and fat fraction of human, goat and bovine milk and are able to enter normal or tumor cells decreasing the expression of the miRNA’s target genes [50]. However, further research is needed to establish whether the goat’s milk used in this study preserved its exosome or bioactive peptide content, and to evaluate if contributed to the observed beneficial effect.

Altogether our results demonstrate that the incorporation of goat’s milk decreased body weight and body fat mass, improved glucose tolerance, prevented adipose tissue hypertrophy and hepatic steatosis in mice fed a HF diet, through an increase in energy expenditure, augmented oxidative fibers in skeletal muscle, and reduced inflammatory markers. Therefore, goat’s milk can be considered a non-pharmacologic strategy to improve the metabolic alterations induced by a HF diet. A summary diagram of the effects in mice metabolism by incorporating dry whole goat milk in a high fat diet is shown in Figure 7.

Finally, to obtain a reference dose for future clinical studies, the body surface area (BSA) normalization method can be used to convert the dose information for mice to equivalent human intake [52,53]. For instance, in the present study consuming a 3.5 g/day of feed containing dry whole goat milk from a conventional diet would provide mice with 0.56 mg/day of bioactive phenolic (Figure 1C, Table 1). This amount of bioactive phenolic for mice of ~25-g body weight in average corresponds to 22.4 mg/kg per day, which multiplied by the animal Km factor (3) and divided by the human Km factor (37) will give a human dose equivalent of 1.81 mg/kg per day. The Km factor is obtained by dividing body weight (kg) and BSA (m^2^) and used for drug dose conversion between mg/kg and mg/m^2^, and its value for different animal species is available in the literature [52]. Accordingly, for an average human adult of 60 kg, the intake of bioactive phenolic from goat milk would correspond to ~108.6 mg/day. Thus, considering, that fresh goat milk contains ~150 mg phenolic/L [10] then the calculated intake would correspond to 0.72 L of fresh milk. This means that the human equivalent dose (~108.6 mg/day) could be supplied by daily consumption of 2.8 glasses/cups of fresh goat milk (250 mL per glass or cup). Furthermore, the source of the feed diet in goats enhances the content of bioactive phenolic in fresh milk as high as 230 mg/L in grazing diet and 300 mg/L in supplemented *Acacia farnesiana* pods diets [10] like those used in the present study. Thus, in these cases the calculated human intake would correspond to only 0.36–0.47 L of fresh milk which is about daily consumption of 1.4–1.8 glasses/cups of fresh goat milk. Alternatively, freeze dried whole goat milk powder corresponding to the calculated human intake of fresh milk could be incorporated as part of a solid diet in similar fashion as described in the present study.

## 4. Materials and Methods

### 4.1. Animals

Male C57BL/6 mice of 5 weeks of age and weighing 20–23 g were obtained from the Experimental Research Department and Animal Care Facility at the Instituto Nacional de Ciencias Médicas y Nutrición Salvador Zubirán (DIEB-INCMNSZ) and housed in micro isolator cages at 23 °C with a 12-h on/12-h off light-dark cycle (7:00 AM–7:00 PM). All animal procedures were conducted in accordance with the recommendations and procedures from the National Institutes of Health guide for care and use of Laboratory Animals [54]. The Animal Care Committee of the Instituto Nacional de Ciencias Médicas y Nutrición Salvador Zubirán (CINVA-INCMNSZ) approved the study (Approval number NAN-1904-18-19-1).

### 4.2. Experimental Diets

Animals were randomly assigned into five groups (*n* = 6) receiving one of the following isoenergetic experimental diets: (1) Control; (2) High-fat (HF); (3) HF + dry whole milk from goats fed a conventional diet (HFCD); (4) HF + dry whole milk from goats fed on grazing (HFG); (5) HF + dry whole milk from goats fed a conventional diet supplemented with 30% of *Acacia farnesiana* pods (HFAF). Goat’s milk was collected and lyophilized according to Delgadillo-Puga et al. [10]. Diets were administered in dry from, and their composition was adjusted according to the recommendations of AIN 93 [55] (Table 1). The study had a duration of twelve weeks where all animals had ad libitum access to water and their respective experimental diet. Body weight was measured once a week and, food intake was measured every other day during the study. Body composition, energy expenditure, glucose and insulin tolerance were evaluated as indicated below at week 10, 11 and 12, respectively as indicated below.

At the end of the study, mice were deprived of food six hours previous to the euthanasia mice were deprived of food. Euthanasia was performed by exposing mice to a lethal dose of sevoflurane (fluoromethyl-2,2,2-trifluoro-1-(trifluoromethyl)ethyl ether). Total blood volume was drawn from the posterior vena cava using a 1 mL syringe coated with heparin. Serum was separated by centrifugation for 10 min at 1800× *g* at 4 °C. Then subcutaneous adipose tissue (SAT), visceral adipose tissue (VAT), brown adipose tissue (BAT), liver, pancreas, skeletal muscle (soleus and gastrocnemius) were rapidly removed and divided in smalls parts. Some samples were frozen in liquid nitrogen and stored at −80 °C. The rest of the samples were fixed in ice-cold 4% (w/v) paraformaldehyde in phosphate buffer saline (PBS) for histological analyses as described below.

### 4.3. Total Phenolic Content and Chemical Composition Analysis of Experimental Diets

Total phenolic content of the diets was determined by the Folin–Ciocalteu colorimetric method described by Singleton et al. [56]. Briefly, experimental diets were diluted in methanol:water (80:20) and extracted according to Delgadillo-Puga et al. [10], Then, 3 mg of diets extracts were diluted in 1 mL of distilled water. Five hundred µL of each stock solution was transferred into a volumetric flask and 3% HCL was added to a final volume of 5 mL. From the resulting solution, an aliquot of 100 µL was mixed with a 2 mL of 2% NaCO_3_ solution and allowed to stand during two minutes. Then, 100 µL of Folin–Ciocalteu reagent (previously diluted 1:1 v/v with water) were added to the solution and incubated at room temperature for 30 min protected from light. Absorbance was measured at 765 nm in a 3.0 mL UV-Quartz cell (Hach Co. cat. 48228-00, Loveland, CO, USA) using a UV spectrophotometric equipment (Beckman, DU-70, Brea, CA, USA). A calibration curve of gallic acid was prepared to estimate the total polyphenols concentration. Results were expressed as mg of gallic acid equivalents (GAE) per 100 g of diet (Table 1).

### 4.4. Serum Determinations

Serum glucose, cholesterol and triacylglycerols (Tg) were determined using a unicel DxC 600 analyzer (Beckman Coulter, Brea, CA, USA). Insulin (cat. 80-INSMS-EO1, ALPCO, Salem, NH, USA) and leptin (cat. KMC228, Thermo Fisher Scientific, Waltham, MA, USA) were determined by ELISA assays according with the manufacturer’s recommendations.

### 4.5. Lipid Extraction, Derivatization and Fatty Acids Quantification

Total lipids from diets and liver were extracted with chloroform-methanol (0.09 g of sample) according to the method described by Folch [57] and the fatty acids then methylated as previously described [58]. The concentration of fatty acids in the diets or liver was determined by gas chromatography (GC) after an organic extraction and its derivatization to methyl esters. The samples were injected into a DB-225 MS 30 m × 0.25 mm × 0.25 µm analytical column (Agilent, Santa Clara, CA, USA) coupled to an auto-sampler (Agilent 6850).

### 4.6. Body Composition and Energy Expenditure Measurement

Body composition (lean and fat mass) was evaluated in each mouse using magnetic resonance (EchoMRI; Echo Medical Systems, Houston, TX, USA). Energy expenditure was measured by indirect calorimetry in an Oxymax Lab Animal Monitoring System (CLAMS; Columbus Instruments, Columbus, OH, USA). Animals were individually housed in Plexiglas cages with an open flow system connected to CLAMS for 24 h. Prior to the test, animals were acclimatized for 24 h, fasted for 12 h in the light period and fed during the dark period. Throughout the test, O_2_ consumption (VO_2_, mL/kg/h or mL/kg lean mass/h) and CO_2_ production (VCO_2_, mL/kg/h) were measured sequentially during 90 s. Multiple linear regression was performed to evaluate the slopes of change of VO_2_ (mL/h) per unit change of body weight or lean mass. The respiratory exchange ratio (RER) was calculated as the average ratio of produced CO_2_ to O_2_ inhaled (VCO_2_/VO_2_).

### 4.7. Intraperitoneal Glucose and Insulin Tolerance Test

The intraperitoneal glucose tolerance test (ipGTT) was performed by an intraperitoneal administration of a glucose load (2 g/kg body weight) after 6 h of fasting. The intraperitoneal insulin tolerance test (ipITT) was performed by an intraperitoneal administration of an insulin load (0.5 UI/kg body weight) after 6 h of fasting. In both test, blood glucose was determined using a blood glucose monitoring system (Freestyle Optium, Abbott Laboratories, Lake Forest, IL, USA) with blood samples collected from the tail vein 0, 15, 30, 45, 60, 90, and 120 min after the glucose administration [59].

### 4.8. Histological Analysis of Liver, Pancreas, White and Brown Adipose Tissue

Samples of liver, pancreas and adipose tissues were fixed in PBS-buffered 4% paraformaldehyde, dehydrated, embedded in paraffin and cut into 4 µm slices. Sections were stained with hematoxylin and eosin (H&E) and observed under a Leica DM750 microscope (Leica, Wetzlar, Germany) using a 20X lens. Analysis of adipocyte area was performed using Adiposoft software as previously described [60]. Measurement of pancreatic islet areas was carried out with ImageJ software (National Institutes of Health, Bethesda, MD, USA). Two different images of each tissue were analyzed after calibration of the software using the scale bar with a length of 100 µm. Perimeter of each islet was drawn manually and total areas were analyzed by each treatment.

### 4.9. Mitochondria Abundance and Lipid Content in Skeletal Muscle

To visualize mitochondrial abundance and lipid content in skeletal muscle; gastrocnemius and soleus muscles were embedded in optimal cutting temperature compound and rapidly frozen by immersion into liquid nitrogen and kept at −80 °C. Muscle succinate dehydrogenase (SDH) activity was determined according to Yamamoto et al. [61] and Kalmar et al. [62]. Briefly, frozen sections (12 μm) were mounted in positively charged slides (Kling-On SFH1103, BIOCARE Medical, Concord, CA, USA) and incubated in SDH staining solution (0.55 mM nitro-blue tetrazolium and 0.05 mM sodium succinate) and incubated at 37 °C for 60 min. Afterwards, the slides were washed with deionized water and sequentially dehydrated (2 min) in 30%, 60% and 90% acetone. Then, the slides were rehydrated (2 min) with 60% and 30% acetone in deionized water. Digital photographs were taken from each section at 20× magnification as described above, and positive fibers (blue) were quantified with Image J software as described previously [63].

To evaluate lipid content in muscle fibers, frozen slides were rapidly fixed by immersion in ice-cold PBS-buffered 4% paraformaldehyde for 10 min, washed in deionized water and incubated for 30 min with the lipophilic dye BODIPY 493/503 (790389, Sigma-Aldrich, 20 µg/mL in PBS). Slides were washed in PBS and mounted with an aqueous mounting medium with DAPI (ProLong Gold Antifade Mountant P36931, Invitrogen, Carlsbad, CA, USA). Digital photographs were taken from each section at 20X magnification under fluorescence illumination. DAPI (nuclei) and BODIPY (fat) images were merged and fat content quantified using ImageJ software.

### 4.10. Lipid Content in Liver

To visualize hepatic neutral lipids, frozen liver samples were sectioned with a cryostat (8 μm) and stained with 0.5% Oil Red O (ORO) in propylene glycol (Sigma-Aldrich). For the quantitative analysis of the ORO staining, images were converted to an 8-bit grayscale in ImageJ according to Mehlem et al. [64] and the integrated density was measured, which is the product of area and mean gray value.

### 4.11. Quantitation of Tumor Necrosis Factor Alpha (TNF-α) in Adipose Tissue

Subcutaneous and visceral adipose tissue punches were homogenized with 500 mL of cold lysis buffer (Merck), centrifuged and stored at −80 °C until analyzed. Content of TNF-α was measured using a 96-well plate of Milliplex MAP kit, with a mouse cytokine/chemokine multiplex magnetic bead panel (Cat. MCYTOMAG-70K, Merck, Kenilworth, NJ, USA). Samples were analyzed in a MAGPIX luminometer and the concentration of TNF-α calculated with a 5-parameter polynomial standard curves ranging from 0.65 to 10,000 pg/mL using the xPonent 4.2 software (Luminex Corporation, Austin, TX, USA). All samples from each treatment were analyzed simultaneously to minimize error.

### 4.12. Immunohistochemistry of UCP-1 in Brown and White Adipose Tissues and Macrophages in Subcutaneous Adipose Tissue

The adaptive thermogenesis marker uncoupling protein one (UCP-1) and macrophage marker F4/80 were determined in 4 μm thick sections of formalin-fixed paraffin embedded tissue according to Leal-Díaz et al. [63] and Méndez-Flores et al. [65] with some modifications. Endogenous peroxidase was blocked with 3% H_2_O_2_ solution. Then, non-specific background staining was avoided with the immunohistochemistry background blocker (Enzo Life Sciences Inc., Farmingdale, NY, USA). Brown and white adipose tissues were incubated with rabbit monoclonal anti-mouse UCP-1 (Abcam, Cambridge, MA, USA) and subcutaneous adipose tissue was incubated with F4/80 IgG2a,k antibody (Santa Cruz Biotechnology, Dallas, TX, USA) diluted at 10 µg/mL for 40 min at room temperature. Binding was identified with Universal Dako labelled streptavidin biotin reagent + peroxidase for primary antibodies from rabbit, mouse and goat (Dako, Glostrup, Denmark). Slides were incubated with streptavidin peroxidase for 15 min, followed by incubation with the peroxidase substrate 3,3′-diaminobenzidine (DAB; SIGMA-Aldrich) for 10 min. The sections were counterstained with hematoxylin, dehydrated with alcohol and xylene, and mounted in resin. Instead of primary antibody, negative controls were performed with normal human serum diluted 1:100 and with the IHC universal negative control reagent specifically designed to work with rabbit, mouse, and goat antibodies (IHC universal negative control reagent, Enzo Life Sciences, Inc.). Instead of the primary antibody the reactive blank was incubated with saline-egg albumin (Sigma-Aldrich). Controls excluded nonspecific staining or endogenous enzymatic activities. We examined at least two different sections of each tissue. Digital photographs were taken from each section at 40X magnification as described above.

### 4.13. Liver, Muscle and Adipose Tissue Gene Expression by Quantitative-Polymerase Chain Reaction (PCR)

(a) Nucleic acid extraction. Total RNA was extracted using a Power SYBR Green Cells-to-Ct kit (Catalog 4402953, Invitrogen). Briefly, frozen tissues were homogenized in ice-cold dissociating reagent in a TissueLyser (Qiagen, Germantown, MD, USA) in order to disintegrate the tissue to release nucleic acids. (b) RNA integrity check. Total RNA was resolved by electrophoresis in a 1.5% agarose gel in 1× borate solution at 100 volts for 50 min. (c) Retrotranscription for cDNA synthesis. Retrotranscription was carried out using the RT Master Mix included in the kit, under the following conditions: incubation at 37 °C for 60 min, then 95 °C for 5 min for inactivation of the RT enzyme. (d) Real time PCR. The analysis was carried out using the LightCycler 480 Instrument II real-time PCR equipment (Roche, Basel, Switzerland) under the following amplification conditions. Enzyme activation: 1 cycle at 95 °C for 10 min, PCR: 55 cycles at 95 °C for 15 s, 60 °C 1 min. Dissociation curve: default configuration of the equipment, finally 1 cooling cycle at 4 °C indefinitely. After Real-time PCR reaction, quantification cycle (Cq) was determined for each sample from the amplification plot. ∆∆Cq value was calculated by subtraction of the 36b4 Cq from each sample Cq and used for data analysis. PCR primers for SREBP1, PPARδ, AMPK, PPARγ2, adiponectin and leptin were synthesized by Sigma-Aldrich. Primers were designed using Primer-BLAST (NCBI, NIH) (Table 2).

### 4.14. Immunoblotting

Tissues were homogenized at 4 °C in ice-cold RIPA buffer containing phosphate-buffered saline (PBS), 1% IGEPAL, 0.5% sodium deoxycholate, 0.1% sodium dodecyl sulphate, 1 mM sodium fluoride, 2 mM sodium orthovanadate, and 1 tablet/10 mL of protease inhibitor mixture (Complete Mini, Roche Diagnostics) in a TissueLyser (Qiagen). The samples were incubated on ice for 30 min, centrifuged at 17,400× *g* for 15 min at 4 °C, and the supernatant was transferred to a new tube and stored at −80 °C until use. Protein concentration was determined with the Lowry method. Protein samples (40 µg) were separated on a 10% SDS-polyacrylamide gel and transferred to a polyvinylidene difluoride (PVDF) membranes (Hybond-P, Amersham, GE Healthcare, Chicago, IL, USA) using a wet electroblotting System (Bio-Rad, Hercules, CA, USA). The membranes were blocked for 1 h with 5% non-fat dry milk, and incubated with primary antibody diluted in blocking solution overnight. Primary antibodies were as follows: AMPKa 1/2 (Santa Cruz Biotechnology, dilution 1:3000 liver and 1:1500 skeletal muscle), p-AMPK, Th2-172 (Santa Cruz Biotechnology, dilution 1:1000 liver and skeletal muscle), JNK (Merck Millipore, Burlington, MA, USA) dilution 1:1000 liver), p-JNK (Thr183/Tyr185, Thr221/Tyr223) (Cell Millipore dilution 1:1000 liver), UCP-1 (abcam dilution 1:1000 brown adipose tissue), GAPDH (Abcam 1:100 brow adipose tissue), HSL (Cell Signaling Danvers, MA, USA, dilution 1:2000 white adipose tissue), p-HSL (Cell Signaling dilution 1:2000 white adipose tissue). The membranes were washed three times with TBS-T for 10 min and then incubated with horseradish peroxidase-conjugated secondary antibody (goat anti-rabbit or rabbit anti-goat 1:3500) for 1.5 h. Visualization was performed using a chemiluminescent detection reagent (Millipore, MA, USA). Digital images of the membranes were obtained by a ChemiDoc MP densitometer and processed by Image Lab software (Bio-Rad). The results are reported as phosphorylated/total protein ratio. A value of 1 was arbitrarily assigned to the control group, which were used as a reference for the other conditions.

### 4.15. Statistical Analyses

Data is expressed as mean ± standard error of the means (S.E.M.). Shaphiro-Wilk normality test was used to check data distribution. All groups were analyzed by one-way ANOVA followed by Tukey multiple comparison post hoc test using GraphPad Prism 7.0 (GraphPad Software, San Diego, CA, USA). The differences were considered statistically significant at *p* < 0.05. Mean values with different lowercase letters show statistical differences between each other.

## 5. Conclusions

The incorporation of goat’s milk to a high-fat diet increased skeletal muscle mass and mitochondrial content, augmented brown adipose tissue thermogenesis and white adipose tissue browning and lipolytic activity. These activities at the molecular level were associated with increased oxygen consumption and energy expenditure, increased in situ lipolysis-mediated thermogenesis in subcutaneous adipose tissue preventing excessive fat mass accretion and adipocyte hypertrophy and consequently, decreasing serum leptin and triglycerides levels. Goat’s milk intake also increased AMPK-mediated lipid oxidation in liver and skeletal muscle and reduced lipogenesis mediated by SREBP-1c in liver, reducing the fat content in both organs; preventing insulin resistance and hepatic steatosis in mice fed with high-fat diet. Dietary goat’s milk also prevented inflammation in liver and macrophage infiltration in adipose tissue. All these beneficial effects of goat’s milk intake were probably caused by the protective proportion of *n*-3/*n*-6 long/chain polyunsaturated fatty acids and the polyphenol content of goat’s milk. Additionally, the supplementation of dairy from goats feed with non-conventional vegetable sources such as AF could increase the nutritional value and bioactive compound content of milk. These molecules could add nutraceutical value to goat’s milk, which could represent a beneficial impact on consumers health.

## Figures and Tables

**Figure 1 ijms-21-05530-f001:**
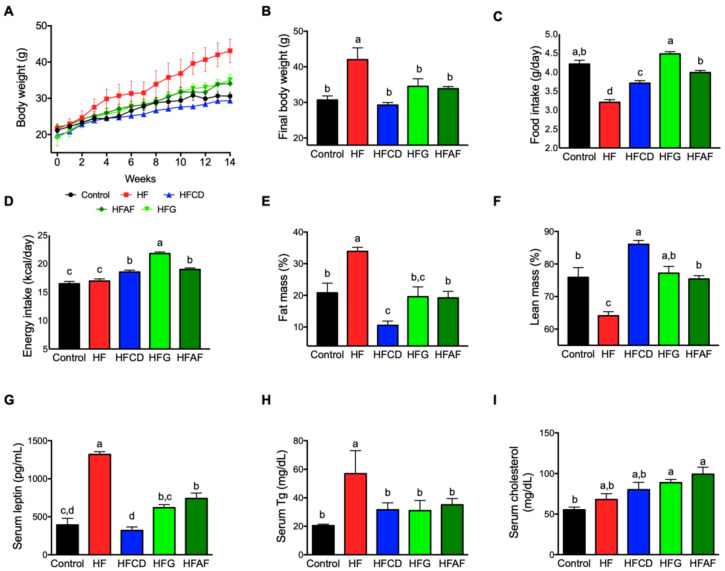
Body weight gain, food and energy intake, body composition and circulating leptin, triglycerides and cholesterol in mice. (**A**) Cumulative body weight throughout the study, (**B**) Final body weight, (**C**) Daily food intake in grams, (**D**) Daily energy intake, (**E**) Fat mass, (**F**) Lean mass (**G**) Serum leptin, (**H**) Serum triglycerides, (**I**) Serum cholesterol of mice fed control diet (control), a high-fat diet (HF) or a high-fat diet with milk from goats fed a conventional diet (HFCD), from goats grazing (HFG) or from goats fed a CD supplemented with *Acacia farnesiana* pods (HFAF). Results are presented as the mean ± S.E.M., *n* = 6 mice per group and analyzed by one-way ANOVA followed by Tukey multiple comparison post hoc test. The differences were considered statistically significant at *p* < 0.05. Mean values with different lowercase letters show statistical differences between each other.

**Figure 2 ijms-21-05530-f002:**
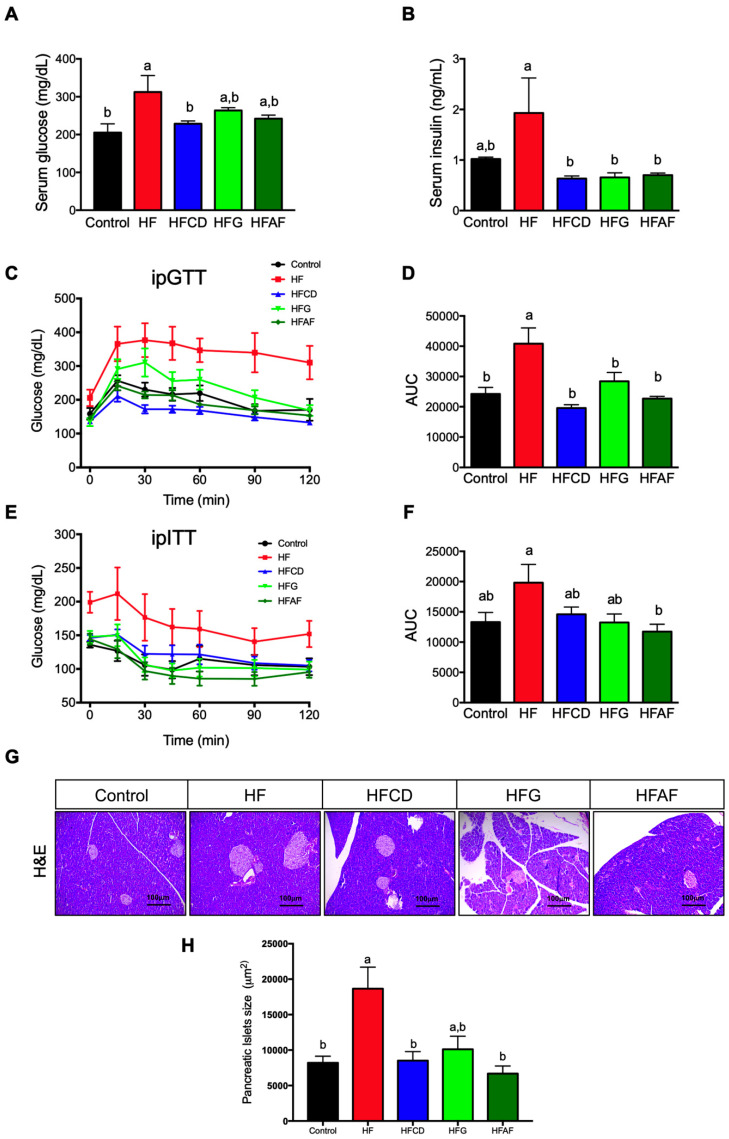
Glucose tolerance, insulin sensitivity and pancreatic islet size of mice. (**A**) Fasting serum glucose, (**B**) Fasting serum insulin, (**C**) Glucose concentrations during intraperitoneal glucose tolerance test (ipGTT), (**D**) ipGTT area under the curve (AUC), (**E**) Glucose concentrations during intraperitoneal insulin tolerance test (ipITT), (**F**) ipITT area under the curve (AUC), (**G**) Representative hematoxylin and eosin stained pancreatic islets (scale bars 100 um), (**H**) Islet size quantification of mice fed control diet (control), a high-fat diet (HF) or a high-fat diet with milk from goats fed a conventional diet (HFCD), from goats grazing (HFG) or from goats fed a CD supplemented with *Acacia farnesiana* pods (HFAF). Results are presented as the mean ± S.E.M., *n* = 6 mice per group and analyzed by one-way ANOVA followed by Tukey multiple comparison post hoc test. The differences were considered statistically significant at *p* < 0.05. Mean values with different lowercase letters show statistical differences between each other.

**Figure 3 ijms-21-05530-f003:**
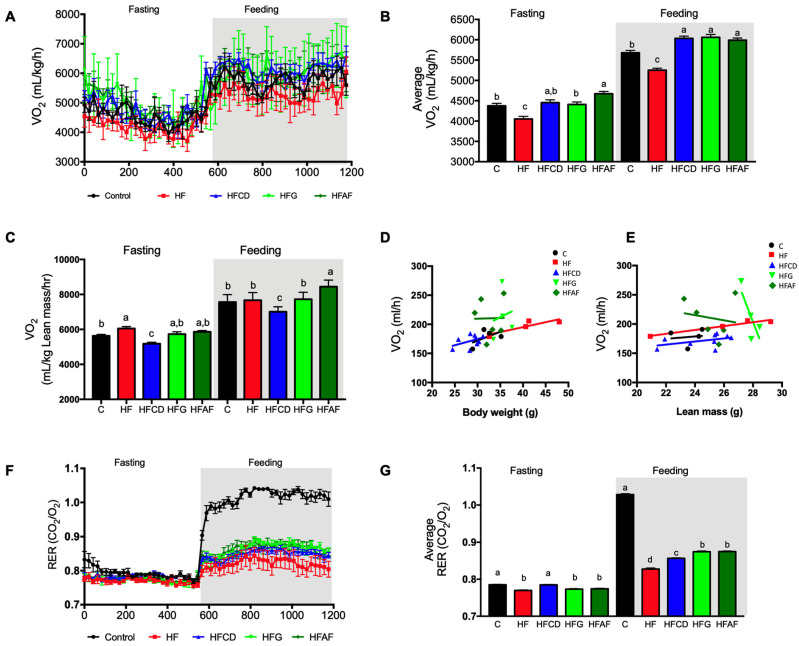
Whole-body energy expenditure and substrate oxidation. (**A**,**B**) Oxygen consumption (VO_2_) normalized to body weight and to (**C**) lean mass, during fasting and feeding periods determined by indirect calorimetry analysis; clear and shaded zones indicate fasting and feeding periods, respectively. (**D**,**E**) Regression plot comparing oxygen consumption (VO_2_) as a function of body weight or lean mass, respectively. (**F**) Respiratory exchange ratio (RER) and (**G**) Average RER during fasting and feeding periods. Indirect calorimetry was performed in mice fed control diet (control), a high-fat diet (HF) or a high-fat diet with milk from goats fed a conventional diet (HFCD), from goats grazing (HFG) or from goats fed a CD supplemented with *Acacia farnesiana* pods (HFAF). Results are presented as the mean ± S.E.M., *n* = 6 mice per group and analyzed by one-way ANOVA followed by Tukey multiple comparison post hoc test. The differences were considered statistically significant at *p* < 0.05. Mean values with different lowercase letters show statistical differences between each other.

**Figure 4 ijms-21-05530-f004:**
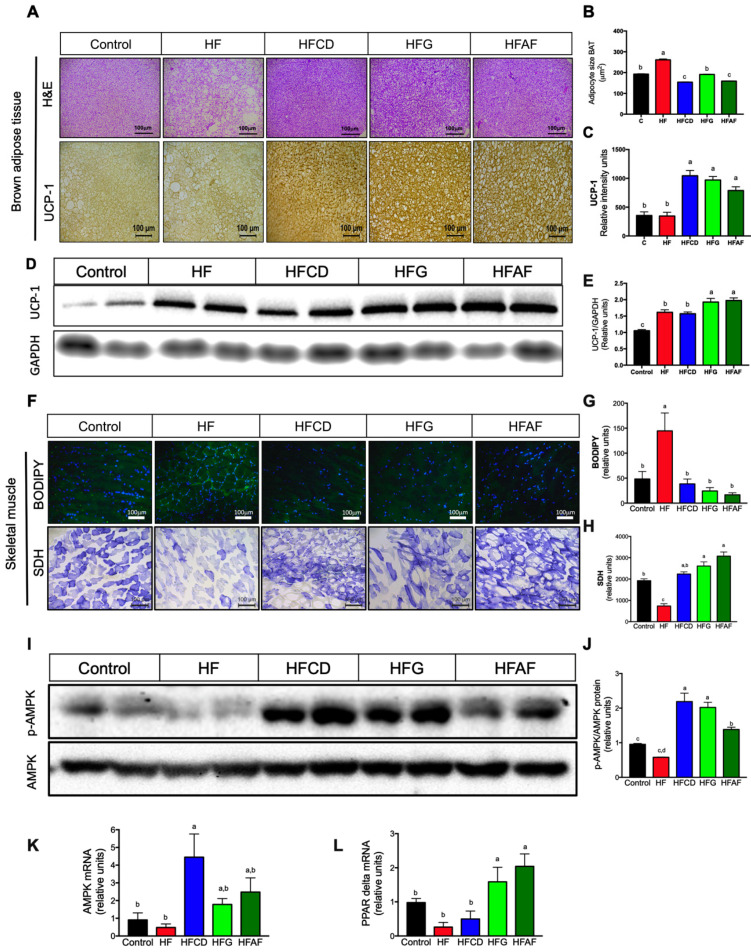
Histological and molecular features of brown adipose tissue and skeletal muscle (**A**) Adipocyte size BAT, (**B**) Uncoupling protein one (UCP-1) immunohistochemistry of BAT, (**C**) UCP-1 immunoblot, (**D**) Densitometric analysis of UCP-1, (**E**) Lipid content (BODIPY staining) and mitochondrial activity (SDH staining) in skeletal muscle, (**F**) Densitometric quantification of BODIPY staining, (**G**) Densitometric quantification of SDH staining, (**H**) Total AMPK and phospho-AMPK immunoblot, (**I**) p-AMPK/total AMPK densitometric analysis, (**J**) AMPK (**K**,**L**) PPARδ mRNA abundance in skeletal muscle of mice fed control diet (control), a high-fat diet (HF) or a high-fat diet with milk from goats fed a conventional diet (HFCD), from goats grazing (HFG) or from goats fed a CD supplemented with *Acacia farnesiana* pods (HFAF). Results are presented as the mean ± S.E.M., *n* = 6 mice per group and analyzed by one-way ANOVA followed by Tukey multiple comparison post hoc test. The differences were considered statistically significant at *p* < 0.05. Mean values with different lowercase letters show statistical differences between each other.

**Figure 5 ijms-21-05530-f005:**
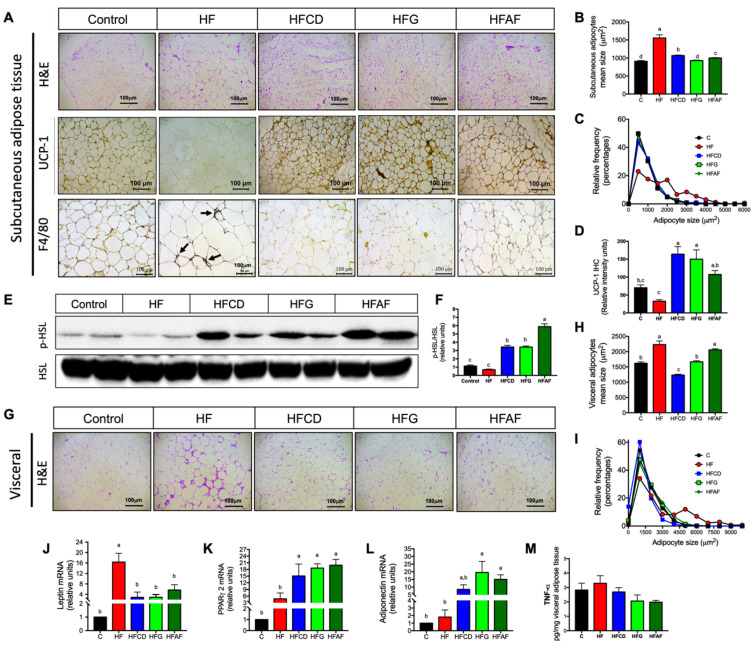
White adipose tissue morphology, macrophage abundance, adipokine gene expression and TNFα content in mice. (**A)** Hematoxylin & eosin staining, uncoupling protein 1 (UCP-1) and macrophage marker F4/80 (arrows) immunohistochemistry in subcutaneous adipose tissue (SAT), (**B**) Mean adipocyte size in SAT, (**C**) Frequency distribution of adipocyte sizes in SAT, (**D**) UCP-1 densitometric analysis, (**E**) Total HSL and phosphorylated HSL immunoblot, (**F**) p-HSL/HSL densitometric analysis, (**G**) Hematoxylin & eosin staining of visceral adipose tissues (VAT), (**H**) Mean adipocyte size of VAT, (**I**) Mean adipocyte size of VAT, subcutaneous adipose tissue mRNA abundance of (**J**) Leptin, (**K**) PPARγ2 and (**L**) Adiponectin, (**M**) TNF-α abundance in adipose tissue homogenates of mice fed control diet (control), a high-fat diet (HF) or a high-fat diet with milk from goats fed a conventional diet (HFCD), from goats grazing (HFG) or from goats fed a CD supplemented with *Acacia farnesiana* pods (HFAF). Results are presented as the mean ± S.E.M., *n* = 6 mice per group and analyzed by one-way ANOVA followed by Tukey multiple comparison post hoc test. The differences were considered statistically significant at *p* < 0.05. Mean values with different lowercase letters show statistical differences between each other.

**Figure 6 ijms-21-05530-f006:**
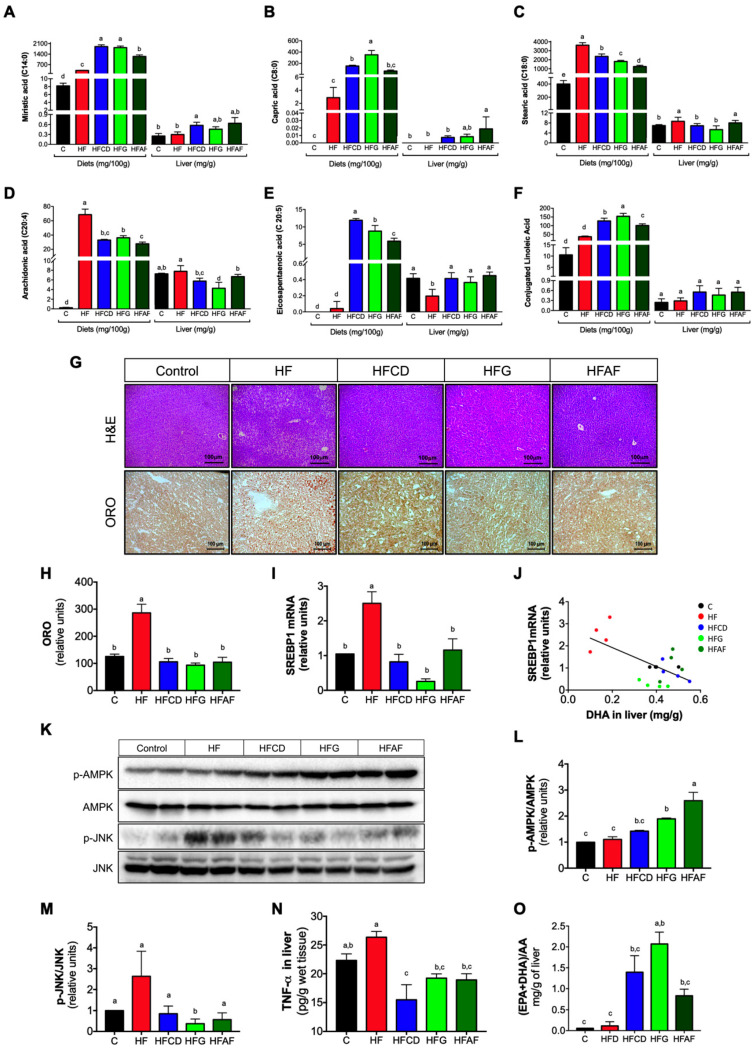
Fatty acid profile of experimental diets and liver; and hepatic lipid content, lipogenic gene expression and inflammatory markers in mice. (**A**) Myristic, (**B**) Capric, (**C**) Stearic, (**D**) Arachidonic, (**E**) Eicosapentaenoic and (**F**) Conjugated linoleic fatty acids content in diets and liver. (**G**) Hematoxylin & eosin and oil red O (ORO) staining in liver sections, (**H**) ORO densitometric quantitation, (**I**) SREBP-1c mRNA abundance. (**J**) Correlation between SREBP-1c mRNA in liver and DHA content in liver. (**K**) Phospho-AMPK, total AMPK, phospho-JNK and total JNK immunoblots. (**L**) Densitometric analysis of phospho-AMPK/AMPK ratio and (**M**) Phospho-JNK/JNK ratio. (**N**) Tumor necrosis factor α (TNF-α) content in liver, and (**O**) EPA+DHA/AA ratio in liver of mice fed control diet (control), a high-fat diet (HF) or a high-fat diet with milk from goats fed a conventional diet (HFCD), from goats grazing (HFG) or from goats fed a CD supplemented with *Acacia farnesiana* pods (HFAF). Results are presented as the mean ± S.E.M., *n* = 6 mice per group and analyzed by one-way ANOVA followed by Tukey multiple comparison post hoc test. The differences were considered statistically significant at *p* < 0.05. Mean values with different lowercase letters show statistical differences between each other.

**Figure 7 ijms-21-05530-f007:**
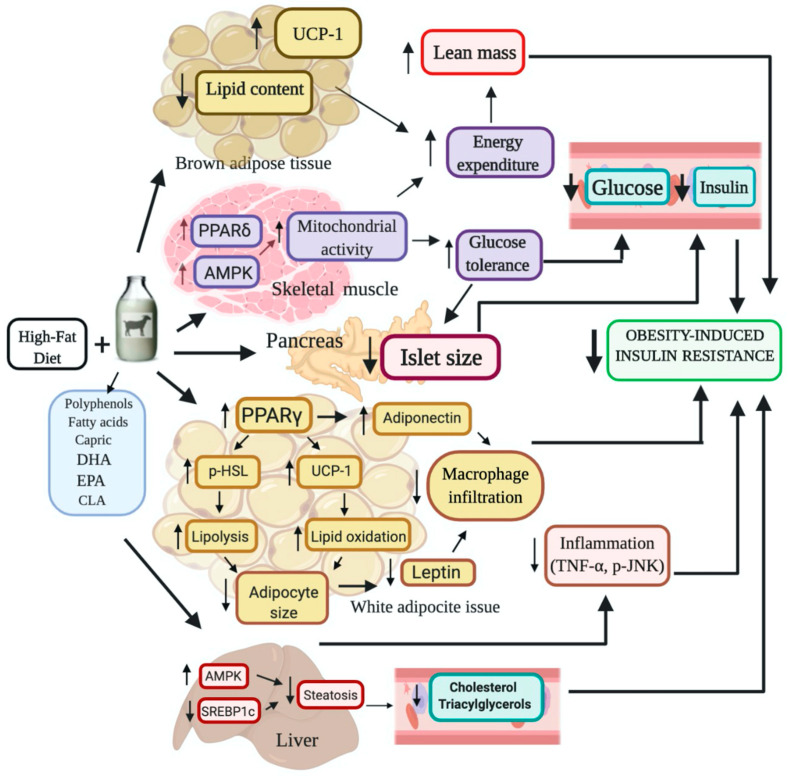
Summary diagram of the effects in mice metabolism by incorporating dry whole goat milk in a high fat DHA = Docosahexaenoic acid; EPA = Eicosapentaenoic acid; CLA = Conjugated linoleic acid; PPARδ = Peroxisome proliferator-activated receptor delta; PPARγ = Peroxisome proliferator-activated receptor gamma; AMPK = Adenine monophosphate (AMP) activated protein kinase; SREBP-1c = Sterol regulatory element binding protein-1c; TNF-α = Tumor necrosis factor alpha; p-JNK = Phosphorylated-Jun N-terminal kinase; UCP-1 = Uncoupling protein 1; HSL = Hormone-sensitive lipase.

**Table 1 ijms-21-05530-t001:** Composition of experimental diets: Control; High-fat (HF); HF + milk from goats fed a conventional diet (HFCD); HF + milk from goats fed on grazing (HFG); HF + milk from goats fed a conventional diet supplemented with 30% of *Acacia farnesiana* pods (HFAF).

Ingredient (%)	Control	HF	HFCD	HFG	HFAF
Dry whole goat milk	--	--	60.0	60.0	60.0
Casein ^a^	20.0	20.0	6.0	1.0	--
Sucrose ^b^	10.0	23.9	12	15.0	14.0
Corn starch ^c^	40.0	--	--	--	--
Maltodextrin ^d^	13.0	13.0	--	--	--
Lard	--	28.0	7.0	9.0	11.0
Soy oil ^e^	7.0	5.0	5.0	5.0	5.0
Cellulose ^f^	5.0	5.0	5.0	5.0	5.0
Vitamin Mix ^g^	1.0	1.1	1.1	1.1	1.1
Mineral Mix ^h^	3.5	3.5	3.5	3.5	3.5
L-Cysteine ^i^	0.3	0.3	0.3	0.3	0.3
Choline ^j^	0.25	0.25	0.25	0.25	0.25
Total	100	100	100	100	100
% Energy (kcal) from:					
Protein	20.6	20.6	20.6	20.8	20.8
Fat	7.0	32.0	32.4	32.0	32.4
Carbohydrates	63.0	36.9	36.5	36.5	36.1
Polyphenol content (mg/GAE per 100 g of diet) ^k^	1.68	1.65	16.1	17.4	17.8

^a^ Casein, vitamin-free test (Envigo Teklad, Indianapolis, IN, USA). ^b^ Sucrose (cane sugar). ^c^ Corn starch (CP ingredients). ^d^ Maltodextrin (MP Biomedicals, Irvine, CA, USA). ^e^ Soy oil (commercial oil). ^f^ Cellulose (AIN Alphacel Non-Nutritive Bulk, MP Biomedicals). ^g^ Vitamin Mix (AIN-76 Vitamin Mix, MP Biomedicals) ^h^ Mineral Mix (AIN-76 Minerals Mix, MP Biomedicals). ^I^ L-Cysteine (Sigma-Aldrich, St. Louis, MO, USA). ^j^ Choline citrate (Sigma-Aldrich). ^k^ Gallic acid equivalents.

**Table 2 ijms-21-05530-t002:** Forward and reverse primers used in real time PCR analysis.

Protein	Gene (Mouse)	Forward Sequence (5′-3′)	Reverse Sequence (3′-5′)
SREBP-1c	*Srebf1*	AGACAAACTGCCCATCCACC	AAGCGGATGTAGTCGATGGC
PPARδ	*Ppard*	CTCTTCATCGCGGCCATCATTCT	TCTGCCATCTTCTGCAGCAGCTT
AMPKα	*PRKAA2*	ACCTGAGAACGTCCTGCTTG	GGCCTGCGTACAATCTTCCT
PPARγ2	*Pparg2*	CTCCTGTTGACCCAGAGCAT	GAAGTTGGTGGGCCAGAA
adiponectin	*Adipoq*	CTGACGACACCAAAAGGGCT	CCAACCTGCACAAGTTCCCT
leptin	*Lep*	CAAGCAGTGCCTATCCAGA	AAGCCCAGGAATGAAGTCCA

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
