# Peer review of "Goat’s Milk Intake Prevents Obesity, Hepatic Steatosis and Insulin Resistance in Mice Fed A High-Fat Diet by Reducing Inflammatory Markers and Increasing Energy Expenditure and Mitochondrial Content in Skeletal Muscle"

_ijms, 2020, doi:10.3390/ijms21155530_

Round 1

Reviewer 1 Report

In this manuscript, the authors examined the effect of goat milk-containing diets on obesity-related parameters in mice fed a high-fat (HF) diet. Overall, goat milk inclusion in diets improved metabolic alterations induced by HF diet. The research strategy and parameters selected were traditional and robust. I found no major flaw in their strategy and experiments.

My biggest concern in the statistical analysis. The authors used Fisher's LSD for multiple comparison throughout the paper (Figs. 1 - 5). This test is a set of individual t tests and does not correct for multiple comparisons. There are many resources, but I cite the GraphPad Prism manual used in this study. If the authors use Tukey test, some significant differences will be removed from the paper. Although I do not consider that the loss of significance will lower the value of their findings, but the description will be changed. This paper should be reviewed in detail after correcting the statistics issue.

https://www.graphpad.com/guides/prism/7/statistics/stat_fishers_lsd.htm

Minor comments
Here are some minor issues need to be fixed in the next revision
L56-L60: "milk and diary products" here mean high-fat products?
Fig. 3E: Can you enlarge the figure? In L175 the authors describe the size of lipid vacuoles of BAT, but it's hard to see them in the photo. Also I would suggest to measure the size of the lipid vacuoles and make a graph like Fig. 4B.
Fig. 4B: Please describe the number of adipocytes and check this data again. If the error bar is SE as descried in M&M, I wonder if you can really find a significant difference between the group.

Author Response

We thank the reviewers for their time revising our manuscript and for their suggestions to improve its quality and clarity.

In this manuscript, the authors examined the effect of goat milk-containing diets on obesity-related parameters in mice fed a high-fat (HF) diet. Overall, goat milk inclusion in diets improved metabolic alterations induced by HF diet. The research strategy and parameters selected were traditional and robust. I found no major flaw in their strategy and experiments.

We thank the reviewer for his/her positive comments and suggestions.

My biggest concern in the statistical analysis. The authors used Fisher's LSD for multiple comparison throughout the paper (Figs. 1 - 5). This test is a set of individual t tests and does not correct for multiple comparisons. There are many resources, but I cite the GraphPad Prism manual used in this study. If the authors use Tukey test, some significant differences will be removed from the paper. Although I do not consider that the loss of significance will lower the value of their findings, but the description will be changed. This paper should be reviewed in detail after correcting the statistics issue.

https://www.graphpad.com/guides/prism/7/statistics/stat_fishers_lsd.htm

We thank the reviewer for his/her suggestion. We have analyzed the data using One-way ANOVA followed by Tukey multiple comparison post hoc test. Most of the significant differences were conserved, except for TNF alpha in visceral adipose tissue (Fig. 5H). We have modified the text accordingly. 

Minor comments

Here are some minor issues need to be fixed in the next revision

L56-L60: "milk and dairy products" here mean high-fat products?

Fig. 3E: Can you enlarge the figure? In L175 the authors describe the size of lipid vacuoles of BAT, but it's hard to see them in the photo. Also I would suggest to measure the size of the lipid vacuoles and make a graph like Fig. 4B.

Fig. 4B: Please describe the number of adipocytes and check this data again. If the error bar is SE as descried in M&M, I wonder if you can really find a significant difference between the group.

We thank the reviewer for his/her suggestions. We meant high-fat milk and dairy products, we have added the specification. We have changed the size of Fig 3E and we have measured the size of the lipid vacuoles (Fig. 4 A, B). Additionally, we have incorporated new results increasing the number of figures, therefore, Fig. 4 is now figure 5. This change can show in the Fig. 5B. We were check and the error bar, and then we found a significant difference between the group.

Reviewer 2 Report

In the current manuscript, Delgadillo-Puga el all argue that the variation on the beneficial effects of milk observed across the different studies might arise from the animal species from which it comes, or the composition of feed. Therefore, the authors propose to evaluate the effect of milk from goat’s fed different diets on the metabolic alterations caused by a high fat diet in mice. Although I found this study valuable due to the clear beneficial effect of goat milk, its current version has some methodological flaws (see point A) as well as some key experiments/measurements missing (see point B), that prevents this manuscript to be publish in its current form.

A) The authors observe an increase in energy intake in the milk supplemented group and a maintenance of the total body weight so they hypothesize that there could be a change in energy expenditure in those groups so they perform indirect calorimetric assays. My main here concern is that they normalize the data to total body weight when the body composition (% lean mass, more oxidative) is clearly different between groups and could completely alter the results.

B) The authors claim that the milk goat supplementation increased brown fat activity because they observe less lipid accumulation but these could be a simple reflection on the lower serum TG (figure 1H). To actually prove a mode active brown adipose tissue they will need to show an increase in uncoupling protein 1 (UCP-1) expression. On that note, brown fat is a very dynamic tissue that can accumulate or burn the TAG deposits very quickly depending on the energy requirements, therefore, to claim browning or whitening of the tissue the authors need to prove that with other molecular and/or metabolic markers (DOI: 10.1007/164_2018_168). 

The same argument applies to the white fat section of the study, could the lack of hypertrophy and the resultant inflammatory response be the result of lower serum TAG? How can they rule that out?

C) On the discussion the authors suggest that the beneficial effects of goat milk could be the result of AMPK activation from the polyphenols present in the milk. However, even though all milk goat’s were equally effective on its positive effects, not all of them seem to be acting thru AMPK signaling (Figure 5F). How can the authors explain that discrepancy?

D) The discussion needs to be shortened and be limited to the discussion and comparison of the results of the current study.

Author Response

We thank the reviewers for their time revising our manuscript and for their suggestions to improve its quality and clarity.

In the current manuscript, Delgadillo-Puga el all argue that the variation on the beneficial effects of milk observed across the different studies might arise from the animal species from which it comes, or the composition of feed. Therefore, the authors propose to evaluate the effect of milk from goat’s fed different diets on the metabolic alterations caused by a high fat diet in mice. Although I found this study valuable due to the clear beneficial effect of goat milk, its current version has some methodological flaws (see point A) as well as some key experiments/measurements missing (see point B), that prevents this manuscript to be publish in its current form.

We thank the reviewer for his/her positive comments and suggestions. 

The authors observe an increase in energy intake in the milk supplemented group and a maintenance of the total body weight so they hypothesize that there could be a change in energy expenditure in those groups so they perform indirect calorimetric assays. My main here concern is that they normalize the data to total body weight when the body composition (% lean mass, more oxidative) is clearly different between groups and could completely alter the results.

We completely agree with the reviewer and apologize for the lack of a thorough analysis before. We have now normalized the data using the lean mass, and in addition performed a multiple regression analysis that is recommended when the body weight and composition are vastly different between groups (Fig. 3      C, D and E).

The authors claim that the milk goat supplementation increased brown fat activity because they observe less lipid accumulation but these could be a simple reflection on the lower serum TG (figure 1H). To actually prove a mode active brown adipose tissue they will need to show an increase in uncoupling protein 1 (UCP-1) expression. On that note, brown fat is a very dynamic tissue that can accumulate or burn the TAG deposits very quickly depending on the energy requirements, therefore, to claim browning or whitening of the tissue the authors need to prove that with other molecular and/or metabolic markers (DOI: 10.1007/164_2018_168).

The same argument applies to the white fat section of the study, could the lack of hypertrophy and the resultant inflammatory response be the result of lower serum TAG? How can they rule that out?

We thank the reviewer for his/her positive comments and suggestions. We evaluated UCP1 expression in  brown adipose tissue by immunohistochemistry and western blot. The results showed that goat `s milk supplementation increased UCP-1 abundance in brown adipose tissue of mice with respect to control and HF. This data is now included in Fig 4.  

To gain insight into the mechanisms trough which subcutaneous adipose tissue of mice fed goat´s milk reduced adipocyte size, we also evaluated UCP-1 by immunohistochemistry in the subcutaneous white adipose tissue along with hormone sensitive lipase phosphorylation state. We found that goat milk supplementation increased UCP-1 expression and p-HSL protein abundance, indicating increased lipolysis of stored triglycerides coupled with increased in situ fatty acid oxidation and thermogenesis. These results suggests that goat´s milk induces white adipose tissue browning and could explain the reduced adipocyte size of adipose tissue of mice fed goat´s milk. This data is now shown in figure 5

The discussion needs to be shortened and be limited to the discussion and comparison of the results of the current study.

This suggestion was attended, the discussion was rewritten to delete some reflections and, in consequence, the references were changed accordingly.     

Round 2

Reviewer 2 Report

The authors adressed in a satisfactory way most of the previous comments therefore I recommend the publication of the manuscript in its current form.